# PROZ Associated with Sorafenib Sensitivity May Serve as a Potential Target to Enhance the Efficacy of Combined Immunotherapy for Hepatocellular Carcinoma

**DOI:** 10.3390/genes13091535

**Published:** 2022-08-26

**Authors:** Yinkui Chen, Xiusheng Qiu, Donghao Wu, Xu Lu, Guanghui Li, Yongsheng Tang, Changchang Jia, Zhiyong Xiong, Tiantian Wang

**Affiliations:** 1Department of Medical Oncology, The Third Affiliated Hospital of Southern Medical University, Guangzhou 510630, China; 2Vaccine Research Institute, The Third Affiliated Hospital of Sun Yat-sen University, Sun Yat-sen University, Guangzhou 510630, China; 3Department of Medical Oncology, The Third Affiliated Hospital of Sun Yat-sen University, Guangzhou 510630, China; 4Department of Hepatic Surgery Liver Transplantation Center, The Third Affiliated Hospital of Sun Yat-sen University, Guangzhou 510630, China; 5Cell-Gene Therapy Translational Medicine Research Center, The Third Affiliated Hospital of Sun Yat-sen University, Guangzhou 510630, China; 6Department of General Surgery, The Third Affiliated Hospital of Sun Yat-sen University, Guangzhou 510630, China

**Keywords:** PROZ, *KDR*, sorafenib, immunotherapy, hepatocellular carcinoma

## Abstract

Targeted combined immunotherapy has significantly improved the prognosis of patients with advanced hepatocellular carcinoma and has now become the primary treatment for advanced hepatocellular carcinoma. However, some patients still have poor efficacy or are resistant to treatment. The further exploration of molecular markers related to efficacy or finding molecular targets to increase efficacy is an urgent problem that needs to be resolved. In this research, we found that *PROZ* was a gene related to *KDR* expression that had significantly low expression in cancer tissue by analyzing the differential genes of cancer tissue and adjacent tissue and the intersection of *KDR*-related genes in hepatocellular carcinoma. The correlation analysis of clinical data showed that the low expression of PROZ was significantly correlated with the poor prognosis of hepatocellular carcinoma, and further studies found that PROZ was closely related to the expression of p-ERK and VEGFR2 in hepatocellular carcinoma. In addition, intracellular detection also showed that the expression of p-ERK increased and VEGFR2 expression decreased after PROZ interference, and PROZ downregulation with increased p-ERK and decreased VEGFR2 was also detected in sorafenib-resistant strains. At the same time, our analysis found that *PROZ* was negatively correlated with genes related to immunotherapy efficacy such as *CD8A*, *CD274* and *GZMA*, and was also negatively correlated with T-cell infiltration in tumor tissue. Conclusion: PROZ is a gene related to the prognosis of hepatocellular carcinoma and it is closely related to the efficacy of sorafenib and immunotherapy. It may serve as a potential molecular target to improve the efficacy of targeted combined immunotherapy.

## 1. Introduction

At present, immune combination therapy based on immune checkpoint inhibitors has achieved good results in the treatment of advanced liver cancer [1,2]. However, there are still many patients who cannot benefit from this combination therapy for various reasons. The insusceptibility of the tumor itself to immunotherapy is the key. By analyzing the mechanism of immune combined with anti-angiogenic drugs, it was found that anti- Vascular endothelial growth factor (VEGF) therapy can reduce the effect of immunosuppression in hepatocellular carcinoma (HCC) and significantly improve the effect of therapy with immune checkpoint inhibitors (ICIs) [3,4]. Therefore, the combination of anti-VEGF and ICIs is an effective method to achieve the synergistic anti-tumor effect. Sorafenib has been the only drug for the treatment of advanced liver cancer for a long time. It is reported that the effective rate of sorafenib in liver cancer is only about 7% [5,6], and it has been downgraded to a second-line drug for advanced liver cancer; however, recent clinical trials have confirmed that the combination of sorafenib and ICIs can also achieve good efficacy [7,8], so the status of sorafenib in the treatment of liver cancer cannot be ignored. In order to better improve the efficacy of anti-VEGF combined with ICIs therapy, we screened the genes that may be related to anti-VEGF therapy through the database, and verified and analyzed these genes, expecting to find some potential combined therapy targets. VEGF receptor 2 (VEGFR2) is the main target of sorafenib and other tyrosine kinase inhibitor (TKI) therapy. We found the target that is positively related to it through a correlation analysis and then performed further analysis. Among them, *PROZ* encodes a liver vitamin-K-dependent glycoprotein that is synthesized in the liver and secreted into the plasma [9]. The liver is the main organ for synthesizing vitamin K. It was also reported that PROZ is related to the prognosis of tumors. In our previous experiments, we found that PROZ was significantly related to sorafenib treatment through the detection of specimens during sorafenib treatment. Some studies have shown that PROZ may be severed as a prognostic marker in liver cancer, but the relationship between PROZ and liver cancer is still unclear. This study further analyzed the correlation between PROZ and the prognosis of hepatocellular carcinoma (HCC), protein-related pathways, and immune indicators, and further explored the role of PROZ in liver cancer prognosis, as well as sorafenib and immunotherapy, which provides a new potential indicator for improving the efficacy of combined immunotherapy for liver cancer.

## 2. Materials and Methods

### 2.1. Tissue Microarrays (TMAs) Construction

We then re-reviewed hematoxylin-and-eosin (H&E)-stained slides and selected the tumor zones to be analyzed in the paraffin-embedded specimens for further TMA design. TMAs were constructed as described previously [10]. Briefly, for each case, two cores were taken from the selected tumor area. First, a hollow needle was utilized to punch the cylindrical tissue cores (1.0 mm in diameter) from the selected donor tissues. Second, the punched tissue was inserted into a recipient paraffin core in a precisely spaced array pattern, using an automatic TMA instrument (Beecher Instruments, Silver Spring, MD, USA). 

### 2.2. Patients

HCC tissue used for TMA construction was derived from 124 consecutive patients diagnosed with HCC by pathological examination, who received surgical resection in the Third Affiliated Hospital of Sun Yat-sen University from September 2003 to September 2010. Patients were included with the following inclusion criteria: pathologically confirmed as HCC; previously without oncological surgery, chemotherapy, and radiotherapy; all patients had the completed follow-up information and paraffin-embedded specimens. This study was approved by the Human Ethics Committee of the Third Affiliated Hospital of Southern Medical University and the Third Affiliated Hospital of Sun Yat-sen University.

### 2.3. Immunohistochemical (IHC)

Tissues were fixed in 10% (*v*/*v*) formaldehyde in PBS, embedded in paraffin, cut into 5 μm sections, and used for H&E staining and IHC staining with specific primary antibodies against p-ERK, PROZ and VEGFR2. Each tissue section (4 µm) was dewaxed, rehydrated, and treated with 0.3% hydrogen peroxide to block endogenous peroxidase, followed by antigen retrieval (sodium citrate, pH 6.0) in microwave. The sections were incubated with rabbit anti-p-ERK (1:100, Cell Signaling Technology, Danvers, MA, USA), rabbit anti-PROZ (1:200, Abcam, Cambridge, UK) and rabbit anti-VEGFR2 (1:100, Abcam) overnight at 4 °C. After washing, the bound antibodies were detected using horseradish-peroxidase-conjugated secondary antibodies (DAKO, Glostrup, Denmark) and diaminobenzidine (DAKO, Denmark), followed by counterstaining with hematoxylin (Keygen Biotech, Nanjing, China). The IHC scoring was reviewed by two pathologists in a double-blind manner. PROZ expression levels were evaluated by integrating the percentage of positive tumor cells and the intensity of positive staining. The staining results were measured semiquantitatively on a scale of 0 (no staining at all), 1 (weak staining), 2 (medium staining), or 3 (strong staining). The staining extent was also scored as 0 (0–10%), 1 (10–24%), 2 (25–50%), 3 (51–75%) or 4 (≥75%). The sum of the intensity and extent scores was considered as the overall IHC score. The rational PROZ IHC cut-off score was selected according to the receiver operating characteristic curve (ROC) analysis as described in our previous study [11]. 

### 2.4. Cell Lines and Cell Culture

Human HCC cells HepG2 were obtained from the American Type Culture Collection (ATCC, Manassas, VA, USA) and maintained in Dulbecco’s modified Eagle’s medium (Thermo Fisher Scientific, Waltham, MA, USA) supplemented with 10% fetal bovine serum (Gibco, Grand Island, NY, USA). The cells were grown at 37 °C in a 5% CO_2_ humidified incubator. Sorafenib-resistant HepG2 cells were cultured with sorafenib for 6 months, the concentration of Sorafenib started at 1 μM and went up to 10 μM. 

### 2.5. Gene Interfering of PROZ in HepG2 Cells Using shRNA

The stable knockdown of target genes was accomplished by lentiviral-based specific short-hairpin RNA (shRNA) delivery. The sequences of the *PROZ* shRNA were as follows: PROZ-sh1-F CCGGGGAACGACATGGACTCCATTCCTCGAGGAATGGAGTCCATGTCGTTCCTTTTTG; PROZ-sh1-R AATTCAAAAAGGAACGACATGGACTCCATTCCTCGAGGAATGGAGTCCATGTCGTTCC; PROZ-sh2-F CCGGGGTACTCACTCTGGTTTAAACCTCGAGGTTTAAACCAGAGTGAGTACCTTTTTG; PROZ-sh2-R AATTCAAAAAGGTACTCACTCTGGTTTAAACCTCGAGGTTTAAACCAGAGTGAGTACC. For the stable knockdown assay, the pLKO.1 vector together with the packing and helper plasmids PSPAX2 and VSVG were co-transfected into HEK-293T cells by lipo3000 (Invitrogen, Carlabad, CA, USA). Viruses were produced, filtered, and titrated according to the instructions, and infected the HepG2 cells with 8 lg/mL polybrene (TR-1003, Sigma, St. louis, MO, USA), followed by screening by puromycin (2 μg/mL). 

### 2.6. Western Blot Analysis

Western blotting was performed as previously described [12]. Briefly, the cells were lysed in sodium dodecyl sulphate (SDS) lysis buffer and the protein concentration was determined using the BCA method. The protein was separated by sodium dodecyl sulphate–polyacrylamide gel electrophoresis (SDS–PAGE), transferred onto nitrocellulose membranes, and incubated overnight at 4 °C with primary antibody. We used primary antibodies against PROZ (1:1000, Abcam), p-ERK (1:1000, Cell Signaling Technology, Danvers, USA), ERK (1:1000, Cell Signaling Technology), VEGFR2 (1:500, Abcam), β-actin (1:2000, Santa Cruz Biotechnology, Dallas, TX, USA). Horseradish-peroxidase-conjugated secondary goat anti-rabbit (1:5000, Cell Signaling Technology) antibodies were used to detect the primary antibodies. Finally, the complex was detected using electrochemiluminescence (Amersham Biosciences, Amersham, UK).

#### 2.6.1. The Cancer Genome Atlas (TCGA) and Gene Expression Omnibus (GEO) Database Analysis

The gene expression data of HCC was downloaded from TCGA (http://portal.gdc.cancer.gov, accessed on 10 June 2020), and the DEGs of HCC and normal tissues in the GSE54238 and GSE14520 datasets were downloaded from the GEO database (https://www.ncbi.nlm.nih.gov/geo, accessed on 10 June 2020). The study was conducted in accordance with the Declaration of Helsinki (as revised in 2013). The correlation between clinical features and hub genes was determined by using the “cor” package in R (http://bioconductor.org/, accessed on 10 June 2020). For the in-depth analysis of the DEGs in HCC, we used the edgeR package in Bioconductor (http://bioconductor.org/, accessed on 10 June 2020) with *p* < 0.05 and |log fold change (FC) >2. All statistical analyses were performed using R software version 4.0 (The R Foundation, Boston, MA, USA).

#### 2.6.2. GO and KEGG Analysis

We conducted Gene Ontology (GO) and Kyoto Encyclopedia of Genes and Genomes (KEGG) enrichment analysis for the *PROZ*-related genes using the R package “Cluster Profiler”, and visualized using the “ggplot2” package.

### 2.7. Statistical Analysis

SPSS for Windows version 21.0 (SPSS, Inc., Chicago, IL, USA) was used for the data analyses. The survival analysis of recurrence-free survival (RFS) or overall survival (OS) was performed using the Kaplan–Meier method, and the LIHC patients were grouped according to the median value of the corresponding variables. The Kaplan–Meier method was used to compare the differences in the survival status between the high- and low- PROZ groups. Moreover, univariable/multivariable Cox proportional hazards analyses were applied to compare the relative prognostic value of the PROZ with that of routine clinicopathological features in the TCGA. All experiments were performed in triplicate; the data were expressed as the means ± standard deviation and analyzed using Student’s *t*-test and Wilcoxon rank sum test. Statistical significance was set at *p* < 0.05. 

## 3. Results

### 3.1. PROZ Was Correlated with KDR Expression

We first calculated the genes that were differentially expressed in HCC in the TCGA database (LIHC) and GEO database (GSE54238, GSE14520), and screened out 419 genes that were differentially expressed in tumors compared to normal tissues (Figure 1A). Since *KDR* (VEGFR-2) is closely related with sorafenib resistance, we counted a total of 381 genes that were positively correlated with *KDR* expression in the TCGA database (LIHC) and GEO database (GSE54238, GSE14520) (Figure 1B). Taking the intersection of the above screened genes, we obtained 129 genes that were differentially expressed in tumors (Figure 1C), of which 37 genes were differentially expressed by logFC > 2 (Figure 1D). Among them, *PROZ* is a vitamin-K-related gene, and the liver is the main organ for vitamin K synthesis. Therefore, we selected the *PROZ* gene for further analysis.

### 3.2. Expression of PROZ in Hepatocellular Carcinoma and Its Correlation with Prognosis of HCC Based on TCGA

First, we analyzed the expression of *PROZ* in HCC tissue and its correlation with the prognosis of HCC through the TCGA database. The results showed that *PROZ* was unrelated to the type of the underlying disease of HCC; there was no difference in the expression of *PROZ* in alcoholic liver disease, Hepatitis B Virus (HBV), Hepatitis B Virus (HCV) and Nonalcoholic fatty liver disease (NAFLD)-related liver cancer (Figure 2A). However, it was highly expressed in adjacent normal liver tissues and significantly lower in cancer tissues. Figure 2B suggests that *PROZ* may be related to the occurrence of tumors. Furthermore, we detected the expression of *PROZ* in different tumor stages according to the T and G stages of HCC, and there was no significant difference in *PROZ* in different T and G stages (Figure 2C,D), suggesting that *PROZ* is not significantly associated with tumor malignancy. We further analyzed the relationship between the expression of *PROZ* and the prognosis of HCC and found that *PROZ* was significantly correlated with the OS of HCC. Then, we divided the expression of PROZ in HCC into two groups, either equally or by the optimal node: both results showed that the overall survival time of the high-*PROZ*-expression patients compared with low *PROZ* expression was significantly prolonged (Figure 2E,F).

### 3.3. The Correlation between the Expression of PROZ in Tissue and Prognosis of HCC Based on Our Database

In order to verify the results of the expression of *PROZ* in HCC and the relationship with prognosis in the TCGA database, we collected tissue from 124 patients with HCC and analyzed the relationship between the expression of PROZ and prognosis in the tissue. The results showed that the expression of PROZ was significantly lower in cancer tissues, while in normal tissues adjacent to the cancer the expression of PROZ was higher (Figure 3A). Further, through tissue microarray detection, we found that the expression of PROZ was significantly different in different liver cancer patients (Figure 3B). Kaplan–Meier analysis showed that the overall survival time of patients with high PROZ expression was significantly increased compared with the low expression group (Figure 3C), and the progression-free survival time was also significantly prolonged (Figure 3D), suggesting that PROZ is closely related to the prognosis of HCC, and patients with high PROZ expression indicate a better prognosis. Further, we analyzed the correlation between PROZ and the patient’s age, gender, liver cirrhosis, hepatitis, vascular invasion, and other indicators (Table 1). The results showed that there was no correlation between PROZ and any of these indicators except tumor size. At the same time, the multivariate analysis indicated that PROZ and vascular invasion were an independent prognostic factor for HCC (Table 2). All these results suggest that PROZ is an independent prognostic factor that can be used as a potential prognostic predictor for HCC, and our results are mostly consistent with the results of the TCGA database analysis (Table 3). Tumor size was negatively correlated with the expression of PROZ in our data, which was different from the results of TCGA. 

### 3.4. Analysis of PROZ-Related Signaling Pathways

In our previous experiments, we found the correlation between PROZ and the prognosis of HCC. Since the function of PROZ in HCC is still unclear, we further analyzed the signaling pathways that *PROZ* may affect in the TCGA database, and the GO analysis revealed that *PROZ* co-expressed genes were mainly related to the organic acid catabolic process, fatty-acid metabolic process, mitochondrial matrix, pyridoxal phosphate binding, and so on (Figure 4A). Through the KEGG pathway analysis, the co-expressed genes were revealed to be fatty-acid degradation, the PPAR signaling pathway, carbon metabolism, butanoate metabolism, and lysine degradation (Figure 4B). The previous analysis also showed that *PROZ* has the highest correlation with the main tumor vascular signals (expression of *KDR*), and the key to the effectiveness of the combined treatment of HCC with ICIs is to block the VEGF pathway of HCC angiogenesis. Sorafenib has a certain inhibitory effect on the VEGFR pathway. Therefore, we further explored the correlation of PROZ with sorafenib treatment.

### 3.5. PROZ Is Associated with Sorafenib Therapy in HCC

Previous studies have found that over-activation of the mitogen-activated protein kinase (MAPK) pathway can lead to tumor resistance to sorafenib. Therefore, in order to verify whether the resistance of sorafenib is related to PROZ, we used the TCGA database to analyze the correlation between the expression of MAPK1 and PROZ, and found that the expression of PROZ was negatively correlated with MAPK1 (Figure 5A) and positively correlated with KDR (Figure 5B). The further detection of HCC tissues indicated that the expression of PROZ and VEGFR2 was significantly lower in tissues with a high expression of p-ERK, while the expression of PROZ and VEGFR2 was higher with a low expression of p-ERK (Figure 5C). However, we found that the expression of p-ERK was increased and the expression of VEGFR2 was decreased in the hepatoma cells after PROZ interference in vitro. Next, we detected sorafenib-resistant strains, and the results showed that p-ERK was significantly overexpressed in sorafenib-resistant strains and VEGFR2 was reduced in the sorafenib-resistant cells, while the expression of PROZ was decreased relative to non-resistant groups (Figure 5D,E and Appendix A).

### 3.6. Potential Relevance of PROZ to Immunotherapy

Studies have reported that the number of CD8-positive T cells in the tumor is significantly positively correlated with the effectiveness of ICIs therapy. In order to detect the correlation between PROZ and ICIs therapy, we analyzed PROZ and genes that reflect CD8 infiltration based on the TCGA database. The results showed that *PROZ* was significantly negatively correlated with *CD8A*, *PD-1*, *GZMA* (Figure 6A–C). Further, we detected the correlation between PROZ and the number of CD3 cells as well as CD8 cells in HCC tissue (Figure 6D), the results confirmed that PROZ was significantly negatively correlated to the number of CD3 and CD8 cells in the HCC tissue. These results suggest the potential relevance of PROZ to immunotherapy.

## 4. Discussion

PROZ’s relationship with tumors: Recent research has shown that PROZ can be used as a marker of pancreatic cancer through protein identification in the serum [13]. At the same time, some studies in HCC have shown through a public database analysis that PROZ is a marker of early-stage HCC, and that a low expression of PROZ is associated with poor prognosis of HCC [14]. Coincidentally, another database-dependent analysis showed that PROZ is also one of the gene markers associated with the prognosis of HCC [15]. These studies both revealed that PROZ may be closely related to the occurrence of tumors and that it can be used as a potential marker for diagnosis and treatment. However, the role of PROZ in tumors, especially in HCC, is still unclear. Through this study, we not only verified the differential expression of PROZ in HCC tissue and adjacent tissue through the TCGA database, but also verified the relationship between PROZ and the prognosis of HCC. Crucially, through randomly collected HCC samples, we not only verified the expression of PROZ in HCC and its relationship with prognosis using the database, but we also found a possible correlation between PROZ and treatment drugs of HCC, which has not been reported so far. We also confirmed the expression of PROZ in sorafenib-resistant strains by in vitro cell experiments, providing direct evidence that PROZ is related to sorafenib. Our results are novel and have very important clinical implications, providing important clues for exploring the function of PROZ in HCC. Studies have found that PROZ is an important hemostasis-related gene, which is mainly related to the anticoagulation process. However, there are few studies that focus on its function [16,17]. The role of PROZ in diseases, especially in tumors, has been less studied. Our study has made a preliminary exploration of the function of PROZ in HCC. Previous studies in hemostasis have found that microRNA may be directly involved in the expression of the PROZ protein, and that PROZ may be a key target of epigenetic regulation [18]. Combined with our findings, as well as the correlation of PROZ with targeted therapy and immunotherapy in HCC, we believe that PROZ may be a target with potential clinical value for sensitizing the drug therapy of HCC.

The relationship between PROZ and the tumor-targeted drug sorafenib: At present, anti-vascular therapy has achieved good efficacy in various tumors [19,20]. For example, bevacizumab significantly prolonged the survival time of non-small-cell lung cancer and achieved very good efficacy in colorectal cancer, too [21,22]. Although anti-vascular therapy in HCC has not achieved as satisfactory efficacy as in lung cancer and colorectal cancer, anti-VEGF therapy represented by sorafenib and lenvatinib used to be the only drug for advanced HCC treatment [5,6,23]. With the advent of immune checkpoint inhibitors, the therapeutic efficacy of various tumors has made rapid progress. However, the efficacy of ICIs alone is not significantly improved for most tumors, while the combination of chemotherapy or targeted therapy is greatly improved the efficacy of antitumor therapy [24,25]. Hepatocellular carcinoma is no exception. Studies have found that ICIs combined with anti-VEGF led to a breakthrough, bringing the treatment of advanced liver cancer into a new stage. However, the biggest problem at present is that it is still difficult for some patients to benefit from the current combination therapy. The reason may be that the patients’ insensitivity to anti-vascular therapy makes the synergistic effect of ICIs combined with anti-VEGF therapy impossible to achieve. From this, we realized that anti-vascular therapy is crucial for immune combination. Previous studies have found that the excessive production of VEGF by the tumor induces the formation of the tumor immunosuppressive microenvironment by promoting the production of immunosuppressive tumor-associated macrophages: Treg cells and MDSC cells. Anti-VEGF therapy can reverse the immunosuppression of the tumor microenvironment, then directly or indirectly increase the infiltration and proliferation of CD8^+^ T cells, thereby significantly enhancing the efficacy of ICIs. Our study found that PROZ may be closely related to tumor angiogenesis by analyzing the pathways of genes closely related to PROZ. Therefore, we speculated that PROZ may be related to the treatment of VEGFR-targeting TKIs, and further analysis found that PROZ was closely related to the activation of the MAPK pathway protein p-ERK that is associated with sorafenib treatment. The relationship between PROZ and sorafenib resistance was further verified in vitro, and it was found that PROZ regulates the resistance of liver cancer cells to sorafenib by affecting the activity of p-ERK. ERK sits at a unique position in the MAPK pathway, as the upstream molecule RAF has very few effectors besides MEK, which has no substrate other than ERK; and ERK is the only activator that is able to stimulate downstream substrates. Phosphorylated ERK (p-ERK) is the activated state, then it translocates to the nucleus and activates transcription factors that regulate different physiological processes [26], such as the ERK-activated genes MYC, NANOG and PHD2, which are the key regulators in drug resistance [27,28,29]. Our study found that the expression of PROZ directly regulates the activity of ERK, combined with the relationship between the efficacy of sorafenib and ERK, suggesting that PROZ may participate in the regulation of cell sensitivity to sorafenib by affecting the activity of ERK. In addition, we also found that PROZ is negatively correlated with the infiltration of CD8^+^ T cells in HCC. These results have not been reported before. There is only one study involved with the relationship between PROZ and immune components based on the Tumor Immune Estimation Resource tool, which found that PROZ has a certain correlation with immune components such as T cells, PD-1, LAG3, TIM3, but there is a lack of histological evidence [30]. The current view is that the enrichment of CD8^+^ T infiltration in the tumor microenvironment is considered to be a reliable basis for the efficacy of immunotherapy; however, the tumor microenvironment in HCC is strongly immunosuppressive. Studies have found that CD8^+^ T tissue infiltration in most liver cancer patients is relatively less; therefore, how to improve CD8^+^ T tissue infiltration is the key to improving the efficacy of immunotherapy [31,32]. Given this background, immunotherapy based on the use of immune checkpoint inhibitors (ICIs), as single agents or in combination with kinase inhibitors, anti-angiogenic drugs, chemotherapeutic agents, or locoregional therapies, offers great promise in the treatment of HCC [33]. Our study found that PROZ was not only significantly correlated with infiltrating CD8^+^ T infiltrates in HCC, but we also confirmed that PROZ intervention had a direct effect on the infiltration of CD8^+^ T cells. Therefore, PROZ may serve as a potential target to increase the tissue infiltration of CD8^+^ T cells. Although our study lacks an in-depth molecular mechanism by which PROZ is involved in regulating MAPK activity and CD8^+^ T infiltration, we provide a direct correlation between them, confirming that changing the level of PROZ can affect sorafenib resistance. Therefore, we believe that our findings are very important; PROZ may become an important potential target to increase the efficacy of targeted immune combination therapy, which provides an important strategy for solving the current difficulties in the treatment of advanced HCC.

## 5. Conclusions

PROZ has been reported to be an important marker for the prognosis of HCC. Our study provides some new findings. PROZ is correlated with sorafenib resistance and the number of CD8-positive T infiltrates that represent effective immunotherapy, providing a new potential therapeutic target to improve the efficacy of targeted immune combination therapy for advanced HCC.

## Figures and Tables

**Figure 1 genes-13-01535-f001:**
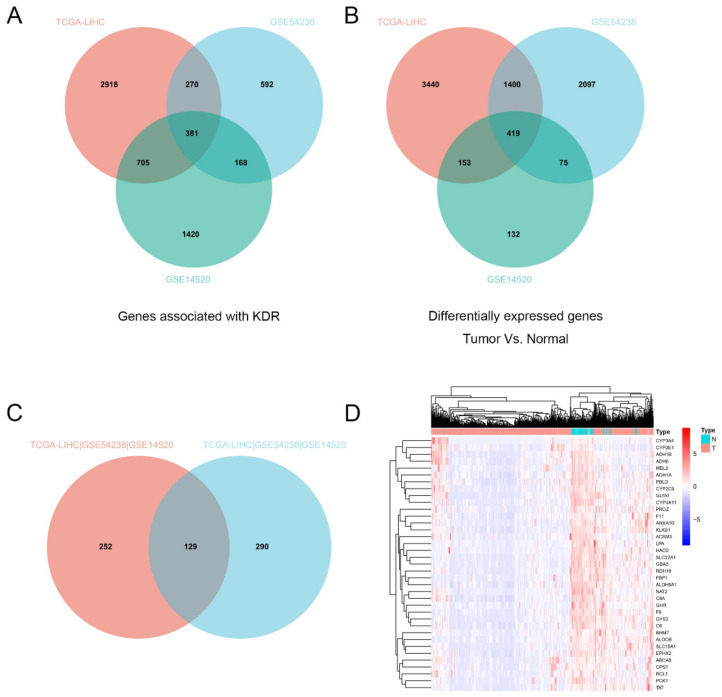
(**A**) Intersection analysis of KDR-associated genes in the TCGA database (LIHC) and GEO database (GSE54238, GSE14520). (**B**) Intersection analysis of differential genes (normal vs. tumor) in TCGA database (LIHC) and GEO database (GSE54238, GSE14520). (**C**) Intersection analysis of *KDR*-associated genes and the differential genes between normal and tumor. (**D**) 37 screened genes were differentially expressed by logFC > 2.

**Figure 2 genes-13-01535-f002:**
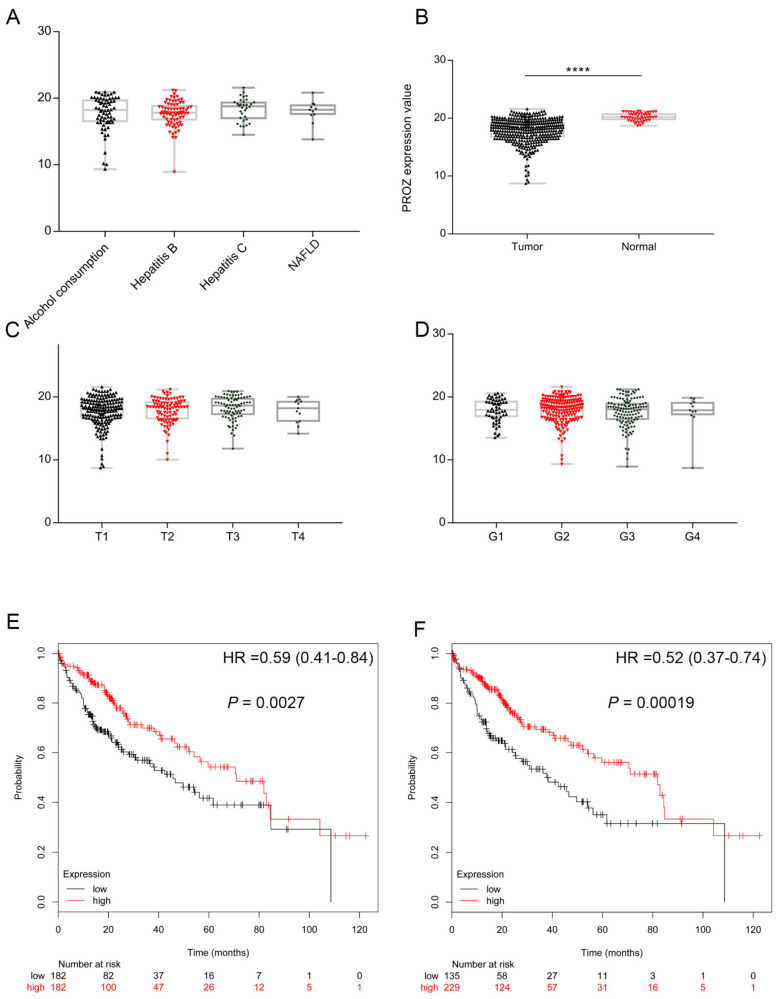
(**A**) Expression of *PROZ* in tumor tissues with different underlying diseases. (**B**) Expression of *PROZ* differed by tumor or normal tissue. Expression of *PROZ* differed in cancer with different T stage (**C**) and histological grade (**D**). Kaplan–Meier curve analysis shows the overall survival in patients with HCC showing high and low *PROZ* expression, the expression of *PROZ* in HCC was divided into two groups equally (**E**) or divided into two groups with the optimal node (**F**) in TCGA cohorts (**** *p* < 0.001).

**Figure 3 genes-13-01535-f003:**
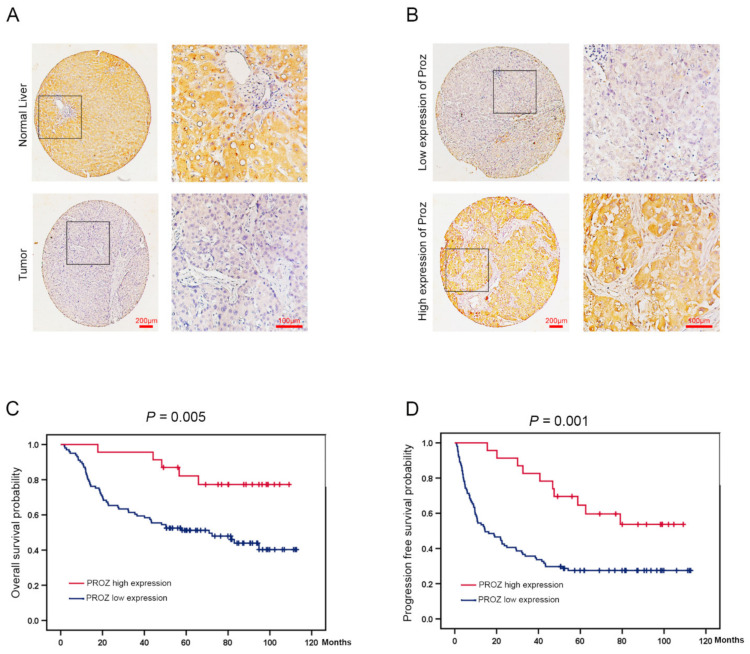
(**A**) PROZ expression was identified by the immunohistochemical (IHC) staining in tumor and normal liver tissues (100×, 400×). (**B**) PROZ expression was identified by the immunohistochemical staining in tumor tissue (100×, 400×). (**C**,**D**) Kaplan–Meier curve analysis shows the OS (*p* = 0.005) and PFS (*p* = 0.001) in patients with HCC showing high and low PROZ expression in tumor.

**Figure 4 genes-13-01535-f004:**
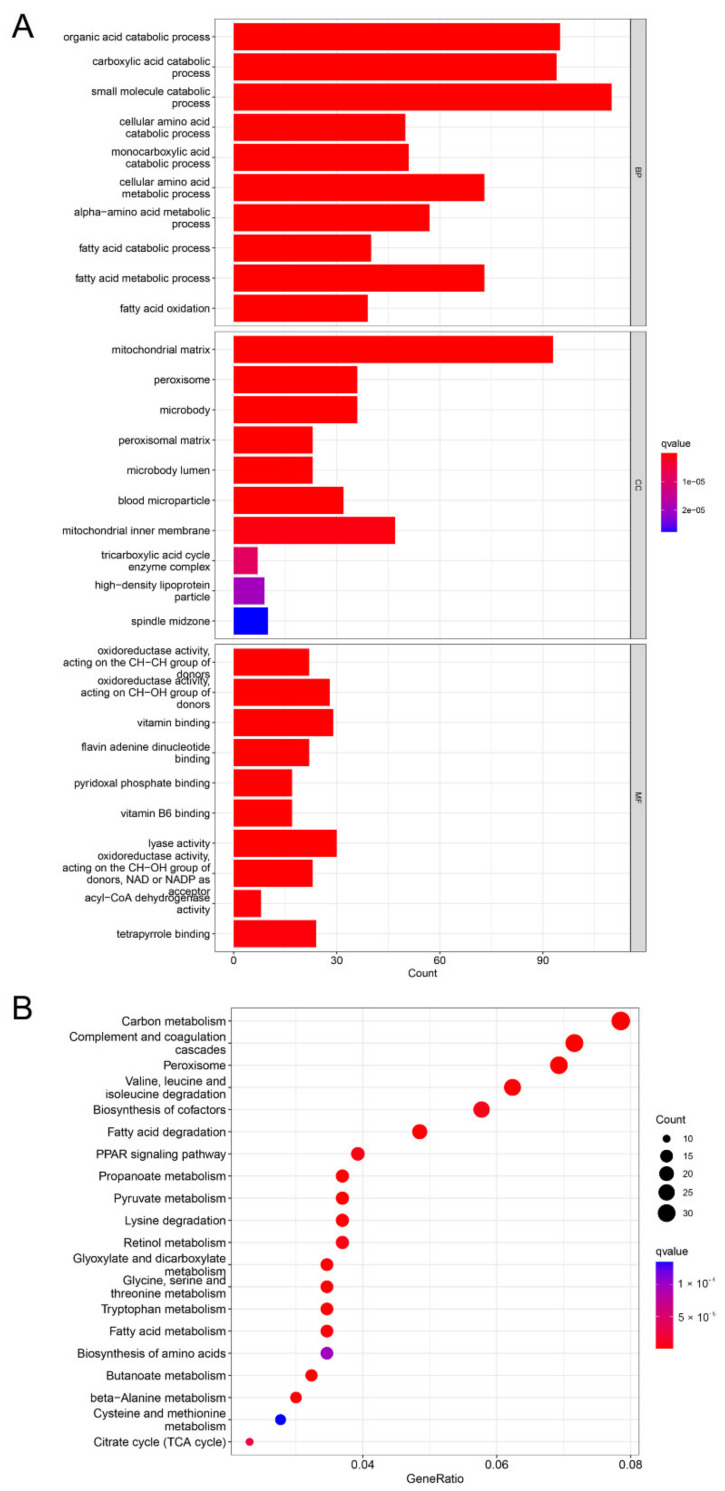
(**A**) KEGG pathway analysis of PROZ-associated genes. (**B**) GO analysis of PROZ-associated genes.

**Figure 5 genes-13-01535-f005:**
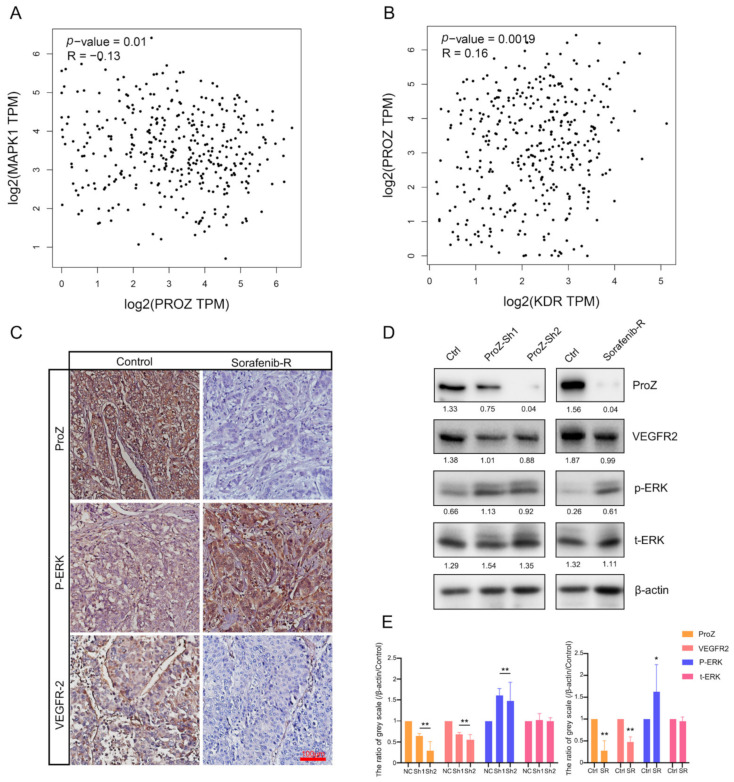
(**A**) Correlation between PROZ gene and MAPK1 gene was analyzed using TCGA data. (**B**) TCGA analysis of the correlation between PROZ and KDR. (**C**) Immunohistochemistry (IHC) staining showed expression changes of p-ERK and VEGFR2 in different expression PROZ of in HCC tumor. (**D**) Expression of p-ERK and VEGFR2 changes after PROZ interference in HCC cells or in sorafenib-resistant HCC cells; the values behind the bands are the ratio of grey scale (target protein compared with β-actin). (**E**) Statistical significance in the Western blot analysis; the grey scale values of the negative control (NC, ShRNA—negative control) and control (Ctrl, the HCC cells were not resistant to sorafenib) were set to 1, the ShRNA–PROZ (Sh1 and Sh2) and sorafenib-resistant cells (SR) were compared with the control group, respectively (Sh1 vs. NC, Sh2 vs. NC and SR vs. Ctrl; * *p* < 0.05; ** *p* < 0.01).

**Figure 6 genes-13-01535-f006:**
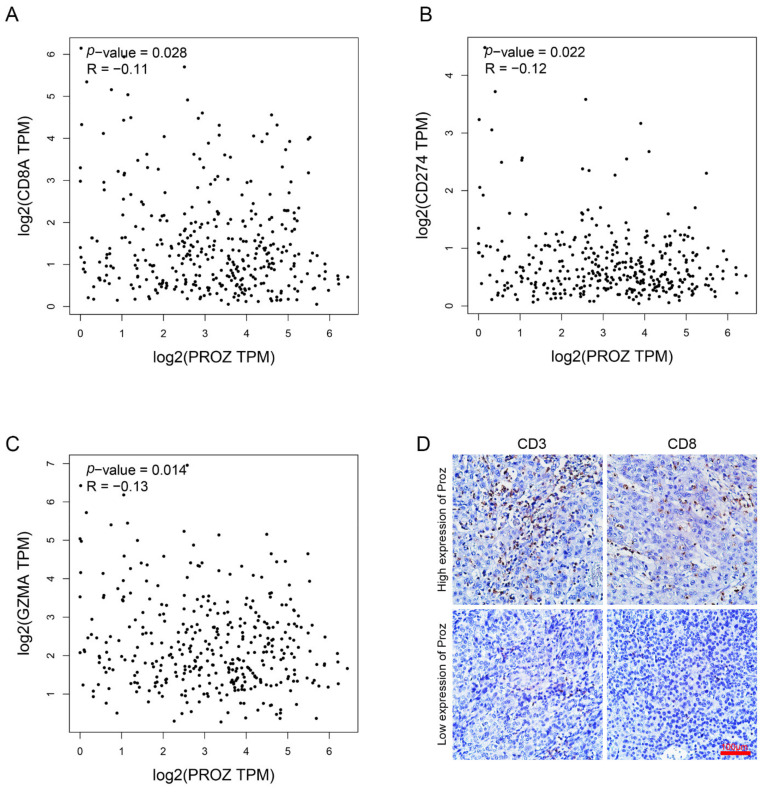
(**A**) Correlation between PROZ gene and CD8A gene was analyzed using TCGA data. (**B**) Correlation between PROZ gene and CD274 gene. (**C**) Correlation between PROZ gene and GZMA gene. (**D**) IHC staining displayed the CD3-positive cells and CD8-positive cells in the tumors expressing high or low PROZ.

**Table 1 genes-13-01535-t001:** PROZ expression status in relation to patient characteristics.

Characteristics	PROZ Positive (*n* = 124)		*p*-Value
	Less	More	
Age (yrs)			0.067
<50	53	7	
≥50	48	16	
Gender			0.208
Male	92	23	
Female	9	0	
Tumor size			**0.020**
≤5 cm	51	18	
>5 cm	50	5	
Vascular invasion			0.241
Positive	22	2	
Negative	79	21	
HBV infection			0.392
Positive	94	20	
Negative	7	3	
Liver cirrhosis			1.000
Negative	16	4	
Positive	85	19	
Tumor number			1.000
>1	26	6	
=1	75	17	
Pathologic stage			0.087
I + II	90	17	
III + IV	11	6	
TNM stage			0.064
1 + 2	47	16	
3 + 4	54	7	
Serum AFP			0.786
<400 ug/L	52	16	
≥400 ug/L	49	7	

HBV, Hepatitis B virus; TNM stage, Tumor Node Metastasis stage.

**Table 2 genes-13-01535-t002:** Multivariate Cox proportional hazards analysis in the overall patients.

Variable	OS		PFS	
	Hazard Ratio (95% CI)	*p*-Value	Hazard Ratio (95% CI)	*p*-Value
Tumor size (<5 cm vs. ≥5 cm)	0.516 (0.225–1.182)	0.117	0.576 (0.293–1.132)	0.109
Vascular invasion (Negative vs. Positive)	0.500 (0.256–0.977)	0.042	0.563 (0.306–1.037)	0.065
TNM stage (1 + 2 vs. 3 + 4)	0.822 (0.329–2.055)	0.676	0.707 (0.335–1.493)	0.364
PROZ expression (High vs. Low)	0.342 (0.135–0.865)	**0.024**	0.385 (0.197–0.754)	**0.005**

**Table 3 genes-13-01535-t003:** PROZ expression status in relation to patient characteristics in TCGA.

Characteristics	PROZ Positive			*p*-Value
	Low	High	Total, *n*	
Age (yrs)			369	0.898
<50	39	38		
≥50	145	146		
Gender			368	0.416
Male	128	120		
Female	57	64		
Lymphocyte infiltration			232	0.204
Absent	52	65		
Mild	48	49		
Severe	12	6		
Vascular invasion			313	0.161
None	107	98		
Micro	41	51		
Macro	11	5		
Neoplasm grade			364	0.383
G1 + G2	112	120		
G3 + G4	70	62		
Pathologic stage			345	0.232
I + II	132	123		
III + IV	40	50		
AFP level (ng/mL)			276	0.296
<400	113	98		
≥400	30	35		
Child-Pugh classification grade			237	0.610
A	110	105		
B + C	10	12		
Liver fibrosis Ishak score			210	0.643
≤4	62	69		
>4	40	39		
Relapse			267	0.437
Yes	44	49		
No	91	83		

## Data Availability

Not applicable.

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
