# Peer review of "PROZ Associated with Sorafenib Sensitivity May Serve as a Potential Target to Enhance the Efficacy of Combined Immunotherapy for Hepatocellular Carcinoma"

_genes, 2022, doi:10.3390/genes13091535_

Round 1

Reviewer 1 Report

Manuscript “PROZ associated with sorafenib sensitivity may serve as a potential target to enhance the efficacy of combined immunotherapy for Hepatocellular carcinoma” submitted by Yin-kui Chen et al have shown that low expression of PROZ significantly correlated with the poor prognosis of hepatocellular carcinoma, impacting overall survival of patients.  Furthermore, this study has shown the close association of PROZ and expression of p-ERK and VEGFR2 in hepatocellular carcinoma.  

Additionally, this study has demonstrated the negative correlation between PROZ and molecular signatures relevant to efficacy of immunotherapy such as CD8A, CD274 and GZMA, and also infiltration of T cells into tumor tissue. The take home message is that PROZ is a gene related to the prognosis of hepatocellular carcinoma and it is closely associated with efficacy of sorafenib treatment and immunotherapeutic approaches in HCC. Authors should discuss the findings of W, Liu Z, Chen W, Jia X, Yin C. “PROZ is a Biomarker for Progression of Early Hepatocellular Carcinoma and Correlated with Tumor-Infiltrating Immune Cell”. Clin Surg. 2022; 7: 3433.

Author Response

Reply:We are very thankful to this kind reminding, we had discussed the finding of W, Liu Z, Chen W, Jia X, Yin C. “PROZ is a Biomarker for Progression of Early Hepatocellular Carcinoma and Correlated with Tumor-Infiltrating Immune Cell”. Clin Surg. 2022; 7: 3433. And added the comments in part of the discussion. “There is only one study involved the relationship between PROZ and immune components based on the Tumor Immune Estimation Resource tool, which found that PROZ has a certain correlation with immune components such as T cells, PD-1, LAG3, TIM3, but lack of histological evidence [26].” (Line 360-363 in page)  

Reviewer 2 Report

Yin-kui Chen et. al. uncovered interesting data regarding therapeutic potential of combined immunotherapy and anti-angiogenesis in HCC.

Point to be discussed:

1.for all Western blot figures, densitometry readings/intensity ratio of each band should be included; the whole Western blot showing all bands and molecular weight markers should be included in the Supplementary Materials; 

2.gene silencing experiments should use at least two gene-specific shRNA

3. Did the author check for hazard's proportionality before proceeding with cox multivariable model?

4. How many IHC staining did the authors analyze and how was the immunohistochemistry quantified in terms of staining positivity? 

5. This reviewer personally misses some insights regarding interesting implications of the authors'findings: As is now well known, tumors grow and evolve through a constant crosstalk with the surrounding microenvironment, and emerging evidence indicates that angiogenesis and immunosuppression frequently occur simultaneously in response to this crosstalk. Accordingly, strategies combining anti-angiogenic therapy and immunotherapy seem to have the potential to tip the balance of the tumor microenvironment and improve treatment response.

In the frame of this thinking HCC is one of most common cancers and the fourth leading cause of death worldwide. Commonly, HCC development occurs in a liver that is severely compromised by chronic injury or inflammation. Liver transplantation, hepatic resection, radiofrequency ablation (RFA), transcatheter arterial chemoembolization (TACE), and targeted therapies based on tyrosine protein kinase inhibitors are the most common treatments. The latter group have been used as the primary choice for a decade. However, tumor microenvironment in HCC is strongly immunosuppressive; thus, new treatment approaches for HCC remain necessary. The great expression of immune checkpoint molecules, such as programmed death-1 (PD-1), cytotoxic T-lymphocyte antigen 4 (CTLA-4), lymphocyte activating gene 3 protein (LAG-3), and mucin domain molecule 3 (TIM-3), on tumor and immune cells and the high levels of immunosuppressive cytokines induce T cell inhibition and represent one of the major mechanisms of HCC immune escape. Recently, immunotherapy based on the use of immune checkpoint inhibitors (ICIs), as single agents or in combination with kinase inhibitors, anti-angiogenic drugs, chemotherapeutic agents, and locoregional therapies, offers great promise in the treatment of HCC (please refer to PMID: 34065489 and expand.

Author Response

To Review2:

1.for all Western blot figures, densitometry readings/intensity ratio of each band should be included; the whole Western blot showing all bands and molecular weight markers should be included in the Supplementary Materials; 

Reply:We are very thankful to this kind reminding, we assessed the densitometry readings/intensity ratio of each band by Image J and the results were added in the Figure 5D and Figure 5E. All bands and molecular weight markers of western blot were provided in the Supplementary Materials. (Figure S1)

2.gene silencing experiments should use at least two gene-specific shRNA

Reply:We are very thankful to this kind reminding, we had used two gene-specific shRNA to silenced the PROZ gene, the results were assessed by Image J again, and we provided the method of gene silencing experiments in the part of method. (Line 122-134 in page 3)

  1. Did the author check for hazard's proportionality before proceeding with cox multivariable model?

Reply:We thank this reviewer greatly for very professional and careful reviewing our manuscript. and we check the hazard's proportionality of variables. the results showed that PROZ, Tumor size, Vascular invasion and TNM stage satisfied the hazard's proportionality in proportional hazard regression mode. So, we corrected the Table 2 with the four variables by using multivariate Cox proportional-hazards analysis.

  1. How many IHC staining did the authors analyze and how was the immunohistochemistry quantified in terms of staining positivity? 

The cases of IHC staining in Figure 3A, Figure 3B and Figure 6D were 124, and there were 6 cases of tissues were tested by IHC staining in each group, including control group and Sorafenib-R group in Figure 5C. the method of immunohistochemistry quantified in terms of staining positivity were provided in the part of IHC stanning method. (Line108-112 in page 3).

  1. This reviewer personally misses some insights regarding interesting implications of the authors'findings: As is now well known, tumors grow and evolve through a constant crosstalk with the surrounding microenvironment, and emerging evidence indicates that angiogenesis and immunosuppression frequently occur simultaneously in response to this crosstalk. Accordingly, strategies combining anti-angiogenic therapy and immunotherapy seem to have the potential to tip the balance of the tumor microenvironment and improve treatment response.

In the frame of this thinking HCC is one of most common cancers and the fourth leading cause of death worldwide. Commonly, HCC development occurs in a liver that is severely compromised by chronic injury or inflammation. Liver transplantation, hepatic resection, radiofrequency ablation (RFA), transcatheter arterial chemoembolization (TACE), and targeted therapies based on tyrosine protein kinase inhibitors are the most common treatments. The latter group have been used as the primary choice for a decade. However, tumor microenvironment in HCC is strongly immunosuppressive; thus, new treatment approaches for HCC remain necessary. The great expression of immune checkpoint molecules, such as programmed death-1 (PD-1), cytotoxic T-lymphocyte antigen 4 (CTLA-4), lymphocyte activating gene 3 protein (LAG-3), and mucin domain molecule 3 (TIM-3), on tumor and immune cells and the high levels of immunosuppressive cytokines induce T cell inhibition and represent one of the major mechanisms of HCC immune escape. Recently, immunotherapy based on the use of immune checkpoint inhibitors (ICIs), as single agents or in combination with kinase inhibitors, anti-angiogenic drugs, chemotherapeutic agents, and locoregional therapies, offers great promise in the treatment of HCC (please refer to PMID: 34065489 and expand.

Reply:We thank this reviewer greatly for very professional suggestion, and added the discussion comments in part of the discussion. “The current view is that the enrichment of CD8+ T infiltration in the tumor microenvironment is considered to be a reliable basis for the efficacy of immunotherapy, however, tumor microenvironment in HCC is strongly immunosuppressive. Studies have found that CD8+T tissue infiltration in most liver cancer patients is relatively less, therefore, how to improve CD8+ T tissue infiltration is the key to improving the efficacy of immunotherapy[31,32]. At this background, immunotherapy based on the use of immune checkpoint inhibitors (ICIs), as single agents or in combination with kinase inhibitors, anti-angiogenic drugs, chemotherapeutic agents, and locoregional therapies, offers great promise in the treatment of HCC[33]. Our study found that PROZ was not only significantly correlated with infiltrating CD8+ T infiltrates in HCC, but also, we confirmed that PROZ intervention had a direct effect on the infiltration of CD8+ T cells. Therefore, PROZ may serve as a potential target to increase tissue infiltration of CD8+T cells.” (Line 382-390 in page 20)

References

  1. Hossain, M.A.; Liu, G.; Dai, B.; Si, Y.; Yang, Q.; Wazir, J.; Birnbaumer, L.; Yang, Y. Reinvigorating exhausted CD8(+) cytotoxic T lymphocytes in the tumor microenvironment and current strategies in cancer immunotherapy. Med. Res. Rev. 2021, 41, 156-201, doi: 10.1002/med.21727.
  2. Farhood, B.; Najafi, M.; Mortezaee, K. CD8(+) cytotoxic T lymphocytes in cancer immunotherapy: A review. J. Cell. Physiol. 2019, 234, 8509-8521, doi: 10.1002/jcp.27782.
  3. Leone, P.; Solimando, A.G.; Fasano, R.; Argentiero, A.; Malerba, E.; Buonavoglia, A.; Lupo, L.G.; De Re, V.; Silvestris, N.; Racanelli, V. The Evolving Role of Immune Checkpoint Inhibitors in Hepatocellular Carcinoma Treatment. Vaccines (Basel) 2021, 9, doi: 10.3390/vaccines9050532.

Reviewer 3 Report

The manuscript entitled as “PROZ associated with sorafenib sensitivity may serve as a potential target to enhance the efficacy of combined immunotherapy for Hepatocellular carcinoma”, is very interesting. The manuscript is well written and designed. However, I think the manuscript can be accepted after a major revision on the following points:

1. I suggest to mention the site of phosphorylation defected in ERK protein, and write is significant in the downstream regulation.

2. Please indicate the scale bar of the representative images.

3. Carefully check the spacing and typo error in the manuscript before the final submission.

4. I strongly recommend to perform the densitometric analysis of the western blots. Because it has to be normalized with the loading control before concluding the results.

5. How many biological replicates have been used in the study in order to make the conclusion?

6. I recommend to perform the statistical significance in the western blot analysis.

Author Response

reviewer 3:

  1. I suggest to mention the site of phosphorylation defected in ERK protein, and write is significant in the downstream regulation.

Reply:This is a professional suggestion for our study. We added the detail of the ERK protein in the part of discussion. “ERK sits at a unique position in the MAPK pathway, as the upstream molecule RAF has very few effectors besides MEK, which has no substrate other than ERK; and ERK is the only activator that is able to stimulate downstream substrates. Phosphorylated ERK (p-ERK) is the activated state, then it translocates to the nucleus and activate transcription factors that regulate different physiological processes[26]. such as ERK activated genes MYC, NANOG and PHD2 were the key regulator in drug resistance[27-29]. Our study found that the expression of PROZ directly regulates the activity of ERK, combined with the relationship between the efficacy of sorafenib and ERK, suggesting that PROZ may participate in the regulation of cell sensitivity to sorafenib by affecting the activity of ERK.” (Line 369-378 in page 20)

References

  1. Liu, F.; Yang, X.; Geng, M.; Huang, M. Targeting ERK, an Achilles' Heel of the MAPK pathway, in cancer therapy. Acta Pharm Sin B 2018, 8, 552-562, doi: 10.1016/j.apsb.2018.01.008.
  2. Li, Z.; Zhou, W.; Zhang, Y.; Sun, W.; Yung, M.; Sun, J.; Li, J.; Chen, C.W.; Li, Z.; Meng, Y.; et al. ERK Regulates HIF1alpha-Mediated Platinum Resistance by Directly Targeting PHD2 in Ovarian Cancer. Clin. Cancer Res. 2019, 25, 5947-5960, doi: 10.1158/1078-0432.CCR-18-4145.
  3. Kim, S.H.; Kim, M.O.; Cho, Y.Y.; Yao, K.; Kim, D.J.; Jeong, C.H.; Yu, D.H.; Bae, K.B.; Cho, E.J.; Jung, S.K.; et al. ERK1 phosphorylates Nanog to regulate protein stability and stem cell self-renewal. Stem Cell Res. 2014, 13, 1-11, doi: 10.1016/j.scr.2014.04.001.
  4. Yeh, E.; Cunningham, M.; Arnold, H.; Chasse, D.; Monteith, T.; Ivaldi, G.; Hahn, W.C.; Stukenberg, P.T.; Shenolikar, S.; Uchida, T.; et al. A signalling pathway controlling c-Myc degradation that impacts oncogenic transformation of human cells. Nat. Cell Biol. 2004, 6, 308-318, doi: 10.1038/ncb1110.

  1. Please indicate the scale bar of the representative images.

Reply:We are sorry for this missing and we added the bar in the new figure according to your kind reminding. (Figure 3, Figure 5 and Figure 6)

  1. Carefully check the spacing and typo error in the manuscript before the final submission.

Reply:We thank this reviewer careful reviewing our manuscript. we checked the spacing and typo error in the manuscript, and corrected the mistakes in our manuscript before the final submission.

  1. I strongly recommend to perform the densitometric analysis of the western blots. Because it has to be normalized with the loading control before concluding the results.

Reply:We are very thankful to this kind reminding, we assessed the densitometry readings/intensity ratio of each band by Image J, the results were added in the Figure 5.

  1. How many biological replicates have been used in the study in order to make the conclusion?

Reply:Biological replicates of IHC staining were mentioned in the 4 question of review 2. “The cases of IHC staining in Figure 3A, Figure 3B and Figure 6D were 124, and there were 6 cases of tissues were tested by IHC staining in each group, including control group and Sorafenib-R group in Figure 5C. the method of immunohistochemistry quantified in terms of staining positivity were provided in the part of IHC stanning method (Line108-112 in page 3)”. The biological replicates of western blot were three times. The gene silencing experiments used at least two gene-specific shRNA.

  1. I recommend to perform the statistical significance in the western blot analysis.

Reply:We had performed the statistical significance in the western blot analysis, and added the results in Figure 5 (Figure 5E).

Round 2

Reviewer 2 Report

I am satisfied with these answers.

Reviewer 3 Report

The manuscript seems better than the previous submission. The manuscript was modified based on the comments given and the response is satisfactory to me.